# Self-Supervised Transformers for fMRI representation

**Itzik Malkiel**[*,1]  **Gony Rosenman**[*,2]  **Lior Wolf**[1]  **Talma Hendler**[2]

[1]`School of Computer Science, Tel Aviv University`
[2]`Sagol School of Neuroscience, Tel Aviv University`

## Abstract

We present TFF, which is a **T**ransformer **f**ramework for the analysis of **f**unctional Magnetic Resonance Imaging (fMRI) data. TFF employs a two-phase training approach. First, self-supervised training is applied to a collection of fMRI scans, where the model is trained to reconstruct 3D volume data. Second, the pre-trained model is fine-tuned on specific tasks, utilizing ground truth labels. Our results show state-of-the-art performance on a variety of fMRI tasks, including age and gender prediction, as well as schizophrenia recognition. Our code for the training, network architecture, and results is attached as supplementary material[1].

**Keywords:** fMRI, Transformers, Self-supervision.

## 1. Introduction

Functional MRI (fMRI) captures the Blood Oxygen Level Dependent (BOLD) signal, which has been shown to be predictive of the diagnosis and characterization of multiple neurological diseases and psychiatric conditions (Zhan and Yu, 2015; Woodward and Cascio, 2015; Xia et al., 2018). fMRI poses a challenge to machine learning, due to the massive amount of signals acquired during a standard scan, which consists of a time series of 3D volumes, as well as the amount of noise that exists in the measurement device, the tradeoff between resolution and inherent noise caused by the patient's motion, repeatability issues due to inter-patient and intra-patient variability, relatively small datasets due to acquisition cost and privacy concerns, and often noisy target labels, which pertain to conditions that are often defined based on a group of symptoms (Kamitani and Tong, 2005).

Multiple advances in deep learning have been applied to fMRI classification, including convolution-based models (Zou et al., 2017; Kawahara et al., 2017), recurrent neural networks (RNN) (Dakka et al., 2017), and graph neural networks (Li et al., 2020). Many of these techniques utilize well-known human brain atlases to parcellate the brain into regions. In our work, we consider the entire, unparcellated volume and apply end-to-end training using a hybrid network that is centered around a transformer (Vaswani et al., 2017) component. Transformers have emerged as a powerful model, which employs multiple attention operations on sequential data. It has become the dominant model in time series forecasting (Li et al., 2019b), natural language understanding (Devlin et al., 2018), and, more recently, computer vision (Dosovitskiy et al., 2020; Li et al., 2019a).

Our network architecture, named TFF, extracts vectors from the raw 3D-volume brain fMRI samples, constructs the vectors as a unified sequence, and propagates them through

---

[*] Contributed equally.
[1] https://github.com/GonyRosenman/TFF

a multi-layer transformer network. The transformer network in TFF allows our model to process, extract and glean information across multiple fMRI frames of the scan. In our research, we fine-tune TFF on multiple datasets, such as the dataset of the Human Connectome Project (Van Essen et al., 2013) and the COBRE[2] and CNP[3] Schizophrenia datasets. We focus on resting-state functional connectomes (van den Heuvel and Hulshoff Pol, 2010; Du et al., 2018) that are known to contain meaningful information in the premise of schizophrenia research (Xia et al., 2018; Woodward and Cascio, 2015), as well as identifying individual fingerprints (Cai et al., 2021).

Our contributions are: (1) we present TFF, which is a novel transformer-based framework that enables transfer learning for fMRI tasks by pre-training on available data. (2) we evaluate and compare our model with other alternatives, reporting state-of-the-art performance for various tasks, including age and gender prediction, and diagnosing Schizophrenia. Importantly, our method operates on the entire fMRI volume and does not require parcellation, which reduces the amount of available data.

**Related Work** Traditional approaches to fMRI data employ a pipeline that first applies a parcellation process to the raw fMRI signal (Arslan et al., 2018). A parcellation procedure aggregates spatially neighboring voxels into local clusters, which represent regions of interest (ROIs). The voxels associated with the same ROIs are averaged and concatenated into a vector, representing a single fMRI frame. Applied to the entire 4D fMRI scan, the parcellation process retrieves multivariate time-series data. Next, given the parcellated data, most techniques infer a functional connectivity (FC) matrix, which is a scalar function that scores the temporal relation between two different regions in the brain. A common FC measure is the Pearson correlation, also known as Static Functional Connectivity (van den Heuvel and Hulshoff Pol, 2010). The FC matrix representing the brain activity, which is also known as the "connectome", has attracted significant interest as a sensitive biomarker for diseases. Finally, supervised machine learning techniques are applied in order to obtain predictions at the level of the individual.

Inspired by the above pipeline, data-driven approaches were applied to obtain a more effective FC function and/or better representations for the parcellated data. Riaz et al. (2020) use 1D convolutions to encode per-region time series, followed by a first multilayer perceptron (MLP) to produce a pairwise similarity matrix and a second MLP to optimize a classification objective. Gadgil et al. (2020) suggest applying Spatial-Temporal Graph Convolutional Networks (ST-GCN) for learning from graph-structured time series data (Yu et al., 2017) and predicting age and gender from parcellated fMRI data. In our work, we evaluate TFF on the same tasks and datasets and obtain superior performance. In addition to being a testbed for comparing multiple algorithms, age prediction is valuable for detecting early signs of brain disease and trajectories of brain degeneration (Chen, 2019; Wang et al., 2019; Churchwell and Yurgelun-Todd, 2013).

## 2. Method

The TFF framework utilizes both 3D convolutional layers and transformer layers, and applies pre-training and a subsequent fine-tuning approach, see Fig. 1. The model utilizes a

---

[2]http://fcon_1000.projects.nitrc.org/indi/retro/cobre.html
[3]https://openneuro.org/datasets/ds000030/versions/00016

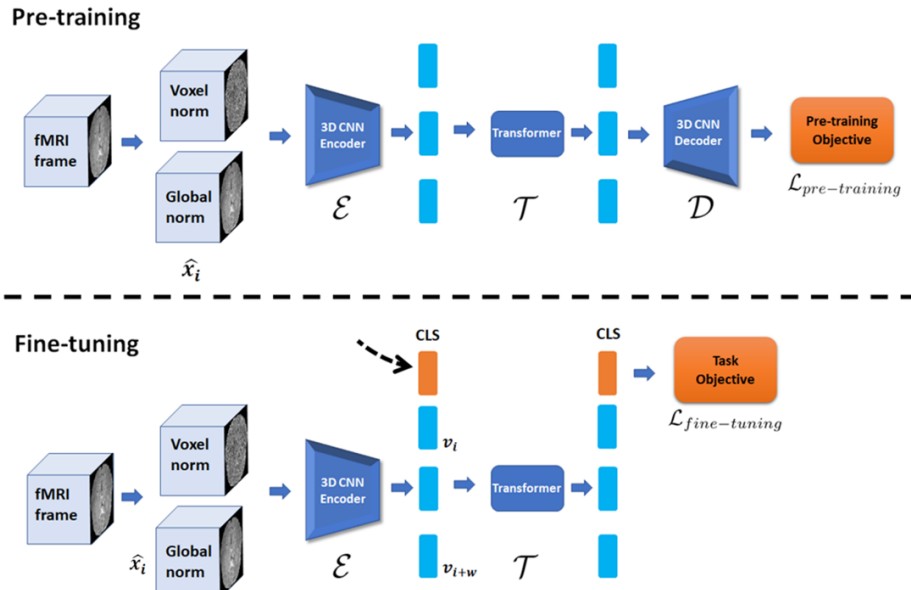

Figure 1: Illustration of the TFF architecture. During pre-training (top), a sequence of fMRI frames are normalized and propagated through $\mathcal{E}$, grouped into a unified sequence and then propagated through the transformer network $\mathcal{T}$. The output sequence is decomposed and propagated through $\mathcal{D}$, which reconstructs the input data. During fine-tuning (bottom), the frames are propagated separately through the pre-trained encoder network, aggregated as a sequence, and a special CLS token is added. The entire sequence is then propagated through the transformer model, which utilizes the embedding of the CLS token for making predictions.

convolution-based encoder $\mathcal{E}$, which operates separately on acquired 3D fMRI frames (each frame is a snapshot of the subject's brain activity at time $t$), mapping each frame into a vector. Next, the model proceeds by aggregating vectors of consecutive frames into a unified sequence and propagating this sequence through a transformer network $\mathcal{T}$.

During pre-training, the transformer output is propagated through a decoder $\mathcal{D}$ that supports self-supervision through reconstructing the original input. During fine-tuning, $\mathcal{D}$ is removed, and $\mathcal{E}$ and $\mathcal{T}$ are optimized directly for the given task, in an end-to-end manner.

**Pre-Training** TFF pre-training employs a two-step training procedure comprised of (i) unsupervised pre-training on unlabeled fMRI scans, and (ii) fine-tuning for a specific supervised task. The first step trains the 3D-convolutional encoder-decoder networks for reconstruction (more details can be found in the architecture section). The auto-encoding strategy with its reconstruction objective allows TFF to learn effective representations for fMRI data.

Given an fMRI scan, with $n$ frames, $X := (x_1, ..., x_n)$, where each $x_i$ is a volumetric data point representing the acquired pulses and echoes in a given interval, $x_i \in R^{W \times H \times D}$ where $W, H, D$ are the width, height, and depth of the acquired data. We first map each frame into two representations by applying two normalization techniques. The first technique

applies *global normalization*, utilizing standard z-score normalization over the entire scan. The second applies *voxel normalization*, which separately z-score normalizes the values of each voxel over the time domain.

The global normalization, denoted by $X^g$, can be expressed as $X^g := \frac{X-\mu}{\sigma}$ where $\mu, \sigma$ are the mean and standard deviation of the entire 4D volume $X$. By focusing on the $qpk$ dimension of a specific frame, the voxel normalization, denoted by $X^v$, can be expressed as $x_{iqpk} = \frac{x_{iqpk} - \mu_{qpk}}{\sigma_{qpk}}$ where $\mu_{qpk}, \sigma_{qpk}$ are the mean and standard deviation of the voxel $qpk$, across all frames in $X$.

Voxel normalization emphasizes the relative activation of a specific voxel in a given interval, while suppressing structural information, see Fig. 4 in the supplementary materials. We denote the concatenation on the channel dimension of the two normalized representations of the entire scan as $\hat{X} := (\hat{x}_1, ..., \hat{x}_n)$.

Next, given $\hat{X}$, we extract a sub-sequence of frames $\hat{x}^w$ with length $w$ and aggregate the frames on the batch dimension. The batches of frames are then propagated through the encoder and the decoder, which outputs data of the same dimension as the input frames. The model is trained to optimize two pixel-wise losses and a perceptual loss, as described below, to reconstruct the global normalized data.

The encoder-decoder architecture imposes a bottleneck with a size of $d$, entailing that each $x_i$ is represented by a single vector $v_i \in \mathcal{R}^d$. Notably, the encoder and decoder networks operate on each frame *separately*, i.e., the convolutions are not applied on batches of frames, and therefore, at this stage, the model cannot extract temporal information.

After the first pre-training stage, we insert the transformer model $\mathcal{T}$ between the two convolution-based encoder and decoder networks and proceed with the same pre-training procedure, using the same reconstruction loss. In this second stage, the transformer architecture enables the model to obtain and process information from the time domain, while training continues in order to optimize the same objective.

In TFF, we adopt a standard transformer architecture (Vaswani et al., 2017). The model first aggregates the vectors provided by $\mathcal{E}$ into a unified sequence and adds a special $CLS$ token to its beginning. The sequence can be denoted by $(CLS, v_i, \ldots, v_{i+w})$, where $w$ is a window hyperparameter dictating the length of input sequences. Next, the sequence is propagated through the $\mathcal{T}$ and its output is propagated through the 3D CNN decoder $\mathcal{D}$. In both pre-training stages, we feed the model with the 2-norm data $\hat{X}$, while training it to only reconstruct globally normalized data.

The two-stage pre-training scheme, chosen over the alternative of a single pre-training phase that optimizes the entire model, stems from the empirical observation that a single training phase is unstable. Additional information can be found in the Experiments section.

**The Pre-Training Losses** During pre-training, we employ two pixel-wise losses and a perceptual loss. The first loss, denoted by $\mathcal{L}_1$, is a standard L1 loss applied between the decoder output and the globally normalized frames $X^g$.

The second loss is an intensity loss, denoted as $\mathcal{L}_1^b$. This loss is based on an L1 applied to a subset of the voxels associated with local intensity values, which are more likely to represent a relevant BOLD signal. More specifically, given a full scan, $(x_1, \ldots, x_n)$, we infer the voxel-normalization $X^v$. Then, for each voxel-normalized frame $x_i^v \in X^v$, we set $x_{i\,qpk}^v = 0$ if $|\hat{x}_{i\,qpk}^v| < b$ and $x_{i\,qpk}^v = \hat{x}_{i\,qpk}^v$ otherwise, where $b$ is a threshold value configured

as the 80% quantile of the absolute of the voxel-normalized values, inside the anatomy and across the sub-sequence. The motivation behind the elimination of the voxels associated with the 80% of the values that are closer to 0 is that these values are typical across many frames, and are therefore unlikely to represent a distinctive signal. An illustration of the intensity loss is depicted in Fig. 5 in the supplementary appendix.

The third loss is a standard perceptual loss, denoted by $\mathcal{L}_p$. Here, we use a pre-trained VGG network (Simonyan and Zisserman, 2014), for which we minimize the L2 distance between the feature maps of the first and second layers extracted from the reconstructed data and input frames. Note that, unlike many perceptual loss terms that focus on high-level features, we focus on low-level features, since our work is in a domain far removed from ImageNet, making high-level features less relevant for this data. To adapt the perceptual loss for 3D data, we construct two batches. The first is the concatenation of the slices of the input data, and the second holds the slices of the reconstructed data. Each batch is propagated separately through the VGG model. The perceptual loss is calculated over the pairs of corresponding slices from each batch and mean-pooled across the pairs.

Finally, the total pre-training objective is: $\mathcal{L}_{pre-training} = \mathcal{L}_1 + \mathcal{L}_1^b + \mathcal{L}_p$.

**Model Architecture**    Our model is composed of a 3D convolution-based encoder, followed by a transformer network. During pre-training, an additional 3D convolutional decoder is used to reconstruct the input data from the transformer output. For the convolution-based encoder and decoder models, we build on the architecture introduced in (Myronenko, 2018).

The transformer architecture is composed of a two-layer multi-head transformer(Vaswani et al., 2017). This network also utilizes a standard positional encoding layer, in which its output is summed with the intermediate vectors (similar to standard language models such as (Devlin et al., 2018)). More details about the network architecture can be found in the supplementary appendix.

**Fine-Tuning**    Fine-tuning involves the $\mathcal{E}$ and $\mathcal{T}$ networks. It optimizes the model with supervision to perform the specific task at hand, by adding a standard classification (regression) head on top of the embedding of the CLS token. The fine-tuning objective is: $\mathcal{L}_{fine-tuning} = -\Sigma_{i=1}^m \mathcal{L}_{cce}\left(y_i, \mathcal{C}\left(\mathcal{T}\left[\mathcal{E}\left(\hat{x}_i^w\right)\right]_0\right)\right)$ where $x_i^w$ is a sub-sequence of frames with length $w$ associated with the label $y_i \in [1...c]$ ($c$ is the number of classes), $\mathcal{C}$ is the classification (or regression) head, $m$ is the number of sub-sequences in the train set, $\mathcal{L}_{cce}$ is a softmax function followed by a standard categorical cross-entropy loss, and 0 is the CLS index. For a binary classification task, we use the same loss, by defining $c = 2$. For regression tasks we replace cross-entropy with a standard MSE.

**Inference**    Given a scan $X$, we infer $\hat{X}$ and extract all sub-sequences of length $w$ and stride $s$. The TFF inference can be written as follows: $\text{TFF}_\mathcal{I} := \frac{\sum_{i=0}^m \mathcal{C}\left(\mathcal{T}\left[\mathcal{E}\left(\left(\hat{X}\right)_{si}^{si+w}\right)\right]_0\right)}{m}$ where $m$ is the number of sub-sequences for the given stride $s$.

## 3. Experiments

We evaluate our model on four tasks and three datasets and compare its performance to three baselines. While it is possible to pre-train our model on multiple datasets, this would hinder our ability to compare it directly with previous work. Therefore, in this study, we pre-train our models separately on each of the given datasets.

Table 1: Gender prediction results on the HCP dataset.

| Model | BAC | Acc. | AUC |
|---|---|---|---|
| TFF | **93.92** | **94.09** | **98.77** |
| TFF$_\text{vanilla}$ | 93.18 | 92.06 | 95.34 |
| ST-GCN | 82.0 | 79.81 | 81.36 |
| Deep-FMRI | 66.91 | 65.45 | 78.0 |

Table 2: Age prediction results on the HCP dataset.

| Model | L1 | L2 | NMSE $\times 10$ |
|---|---|---|---|
| TFF | **2.73** | **10.93** | **0.14** |
| TFF$_\text{vanilla}$ | 3.21 | 14.13 | 0.19 |
| ST-GCN | 3.16 | 13.53 | 0.18 |
| Deep-fMRI | 3.48 | 17.59 | 0.25 |

Table 3: Schizophrenia classification results on the COBRE dataset.

| Model | Accuracy | AUC | BAC |
|---|---|---|---|
| TFF | **70.0** | **68.3** | **69.2** |
| TFF$_\text{vanilla}$ | 43.3 | 44.2 | 44.2 |
| ST-GCN | 57.2 | 64.3 | 58.6 |
| Deep-fMRI | 68.3 | 62.0 | 65.6 |

**The Datasets**    *The Human Connectome Project (HCP)* is a collection of publicly available functional MRI scans (Van Essen et al., 2013). The dataset contains 1095 scans of different subjects, 595 females and 500 males. The age of the subjects ranges between 22 and 36. Each scan includes 1200 fMRI frames, acquired while subjects were in a resting state. In our study, we focus on predicting age (regression task) and gender (binary classification) from fMRI scans. These tasks can help shed light on the relationship between brain activity, age, and gender, especially in the context of neuropsychiatric research. *The Center for Biomedical Research Excellence (COBRE)* is a dataset containing resting-state functional MRI data of 75 healthy control patients along with 72 schizophrenia patients.Given the fMRI scans, the task we consider is to predict whether a subject is healthy or should be diagnosed with schizophrenia (binary classification). *Consortium for Neuropsychiatric Phenomics (CNP)* is an fMRI dataset acquired as part of the UCLA Consortium for Neuropsychiatric Phenomics LA5c Study(Gorgolewski et al., 2017)[4]. The dataset incorporates resting-state fMRI scans of 130 healthy controls subjects and 50 subjects diagnosed with schizophrenia. Here, too, the task we consider is binary schizophrenia classification based on the acquired fMRI scans.

The exact train-validation-test splits, and more information about all datasets can be found in the supplementary appendix.

**The Baselines**    *Spatial-Temporal Graph Convolutional Networks (ST-GCN)* is a computer vision technique for learning from graph-structured time series data (Yu et al., 2017) that was recently shown to produce state-of-the-art results for age and gender prediction from fMRI scans (Gadgil et al., 2020). In this baseline, the fMRI data is parcellated, normalized and fed into the ST-GCN model. *Deep-fMRI* is an end-to-end deep learning architecture for classifying pathologies in fMRI data(Riaz et al., 2020). It receives parcellated fMRI signals as input and outputs a diagnosis. The model is composed of three components. The first component applies a convolution-based network to extract features from the input scan, outputting a vector for each brain region. The second component is a network that operates on all pairs of brain regions, by concatenating the vectors and propagating them through an MLP regression layer. The network then predicts a correlation matrix for each region pair. The last component is an MLP classification network, which makes predictions based on the estimated correlation matrix. Riaz et al. (2020) report that Deep-fMRI outperforms other alternatives, such as correlation-based functional connectivity, clustering-based technique, as well as the FCNet model (Riaz et al., 2017), which is another convolution-based model. $TFF_{vanilla}$ is the TFF model initialized from scratch, without the pre-training procedure.

---

[4]https://openneuro.org/datasets/ds000030/versions/00016

Table 4: Schizophrenia classification results on the CNP dataset.

| Model | Accuracy | AUC | BAC |
|---|---|---|---|
| TFF | **88.2** | **90.0** | **87.9** |
| TFF$_{\text{vanilla}}$ | 58.8 | 52.9 | 50.0 |
| ST-GCN | 82.3 | 87.1 | 82.8 |
| Deep-fMRI | 76.5 | 84.3 | 77.9 |

Table 5: Ablation study, see text for variants (i)–(v).

| | Age pred. | | | Gender pred. | | |
|---|---|---|---|---|---|---|
| | L1 | L2 | NMSE | Acc. | BAC | AUC |
| (i) no intensity loss | 2.96 | 11.71 | 0.16 | 93.77 | 92.95 | 96.06 |
| (ii) no L1 loss | 3.12 | 13.07 | 0.18 | 89.66 | 87.43 | 91.25 |
| (iii) no perceptual loss | 3.02 | 12.11 | 0.17 | 93.47 | 93.02 | 96.86 |
| (iv) no 2-norm | 3.09 | 12.65 | 0.17 | 93.07 | 92.96 | 93.37 |
| (v) one-step pre-training | 3.21 | 14.33 | 0.19 | 89.97 | 91.02 | 90.38 |
| Full method | **2.73** | **10.93** | **0.14** | **94.09** | **93.92** | **98.77** |

**Metrics** Accuracy, balanced accuracy (BAC), and Area Under the Receiver Operating Characteristic curve (AUROC) were measured for classification. For the regression task, we report the L1, L2, and the Normalized Mean Square Error (NMSE) metrics defined as $NMSE(\hat{y}, y) = \frac{\text{MSE}(\hat{y},y)}{\text{MSE}(y,0)}$, where $\hat{y}, y$ are predicted and ground truth values, respectively.

**Implementation details** TFF utilizes the AdamW (Loshchilov and Hutter, 2017) optimizer, with a weight decay of 0.01. The window size is set to $w = 20$, with a stride of $s = 10$. The encoder architecture imposes an intermediate feature vector of size $d = 2640$. In our experiments, all TFF pre-training and fine-tuning used a single GPU (either V100 or Titan X), and each single-step training procedure ran for less than 24 hours, with a standard early stopping strategy. In most cases, the cumulative time of the two-step pre-training is less than 28 hours, and fine-tuning converges within 5-18 hours (depending on the task).

**Results** Tab. 1 presents the results of all models evaluated on the HCP dataset for the gender prediction task. The TFF model was pre-trained on the HCP dataset, according to our proposed training and objective. All models were fine-tuned with supervision, utilizing the gender labels available for each subject in the HCP dataset. As can be be seen in the table, TFF outperforms the alternatives by a sizeable margin. Specifically, compared to the ST-GCN model, which was previously considered state-of-the-art for this task, our model yields an improvement of more than 10% in BAC. Interestingly, we observe that the full TFF method also outperforms the TFF$_{\text{vanilla}}$ baseline by ∼0.8, ∼2 and ∼3.4 points of BAC, accuracy and AUC, respectively.

Table 2 depicts the performance of all models evaluated on the age prediction task from the HCP dataset. This task is formulated as a regression task, where the models are expected to predict the exact age of each subject. Each of the models in this evaluation applies a standard regression head and optimizes an L1 loss w.r.t the ground truth age labels. As can be seen in the table, TFF shows a clear advantage over the other techniques. Compared to ST-GCN, the second best model in this evaluation, TFF yields an absolute improvement of ∼0.4, ∼2.6 on L1 and L2 respectively, and a relative improvement of 23% in NMSE. Interestingly, the gap in performance is even larger with respect to the TFF$_{\text{vanilla}}$ baseline, for which TFF improves in absolute scores of ∼0.4, ∼3.2 for L1 and L2, and a relative improvement of ∼26% in NMSE. This can be attributed to the importance of the pre-training procedure, which allows TFF to learn an effective representation for 4D fMRI data prior to the fine-tuning procedure. Statistical significance for the age prediction task can be found in Appendix H.

We further evaluate our models on two datasets for pathological classification. Tab. 3 presents the results for schizophrenia classification on the COBRE dataset. As can be seen in the table, TFF outperforms the alternatives by a sizeable margin. Looking at the AUC score, TFF outperforms ST-GCN and DeepfMRI by 4 and 8.3 points, respectively. Interestingly, by neglecting the pre-training procedure, the TFF$_\text{vanilla}$ fails to converge on the task. This can be attributed to (1) the relatively smaller number of samples, which hinders the ability of the TFF$_\text{vanilla}$ model to generalize correctly in the case of unseen samples, and (2) to the importance of the pre-training procedure for extracting valuable features in advance, which is crucial to the convergence of the fine-tuning procedure.

Tab. 4 presents the evaluation results of the various models on the CNP dataset. As can be seen, the TFF model greatly outperforms the alternatives. Specifically, TFF achieves a BCA score of 87.9, an improvement of more than 5.1 and 10 absolute points, respectively, over the ST-GCN and Deep-fMRI baselines. Looking at the TFF$_\text{vanilla}$ baseline, we observe that the model suffers from poor performance.

Comparing the convergence of TFF and TFF$_\text{vanilla}$ during the shared fine-tuning stage (appendix Fig. 7), it is evident that TFF$_\text{vanilla}$ struggles to converge, while the full model produces better results across the entire course of training. We attribute the enhanced performance of TFF to the effectiveness of the pre-training procedure. Appendix G presents further empirical evidence in support of pre-training, in a different domain of MRI classification.

**Variable Training Set Size** We vary the amount of data available for training and evaluate the performance of each model for schizophrenia classification on the CNP dataset. The results, as shown in Fig. 8 in the appendix, indicate that the performance of all four methods drops significantly when the amount of training data is reduced. It is also evident that for all training set sizes the full TFF method is better than the baselines.

**Ablation Study** Table 5 presents an ablation study for TFF on the age prediction and gender prediction tasks. The following variants are considered: (i) TFF without intensity loss. (ii) TFF without L1 loss. (iii) no perceptual loss. (iv) training TFF solely on the global normalization data (i.e. neglecting the voxel-normalization input). (v) TFF with one-step pre-training, which trains all three networks $\mathcal{E}, \mathcal{T}, \mathcal{D}$ in one phase. Different from the full method, this model optimizes the transformer weights on top of a randomly initialized encoder $\mathcal{E}$. The results indicate that it is highly beneficial to employ all three losses during the pre-training procedure, that voxel-normalization is crucial for model convergence, and that two-step pre-training is a better alternative to a single step.

## 4. Conclusions

TFF is a novel framework for the analysis of fMRI data. It considers the entire 4D fMRI volume data and applies end-to-end training, using a transformer-based architecture. TFF training consists of two phases, a self-supervised pre-training procedure and subsequent fine-tuning, which optimizes the model for a specific task. Importantly, the pre-training procedure was found to be crucial for improved accuracy. Our experiments demonstrate state-of-the-art performance on a variety of fMRI tasks, including age and gender prediction, as well as schizophrenia recognition. One of the properties of TFF, which was not considered in this work, is that it can be pre-trained on a large amount of unlabeled data, and fine-tuned on relatively smaller datasets, which are common in the field of medical imaging.

**Acknowledgments** This project has received funding from the European Research Council (ERC) under the European Union's Horizon 2020 research and innovation programme (grant ERC CoG 725974). The contribution of I.M. is part of PhD thesis research conducted at Tel Aviv University. This work was additionally supported by the ISRAEL SCIENCE FOUNDATION (grant No. 2923/20), within the Israel Precision Medicine Partnership program.

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

# Supplementary Appendices

## Appendix A. More Details About the Pre-Training

The voxel normalization emphasizes the temporal activations of specific voxels in a given sequence and suppresses the structural information in the acquired scan. A representative sample of the voxel normalization along with the global normalization of the same slice can be seen in Fig. 2

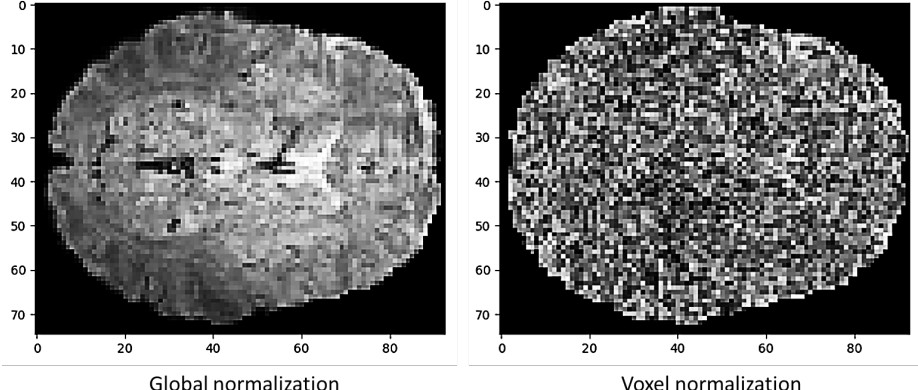

Global normalization                    Voxel normalization

Figure 2: A representative 2D slice from an acquired 3D fMRI frame, applied through global normalization (left) and voxel normalization (right). Global normalization applies the same scaling and shifts to all voxels, yielding a volume that is visually similar to the original volume. Voxel normalization scales and shifts each voxel separately, by looking at its values across the entire scan. The resulting voxel-normalized data suppresses structural information and emphasizes voxels associated with values that are far from their mean value.

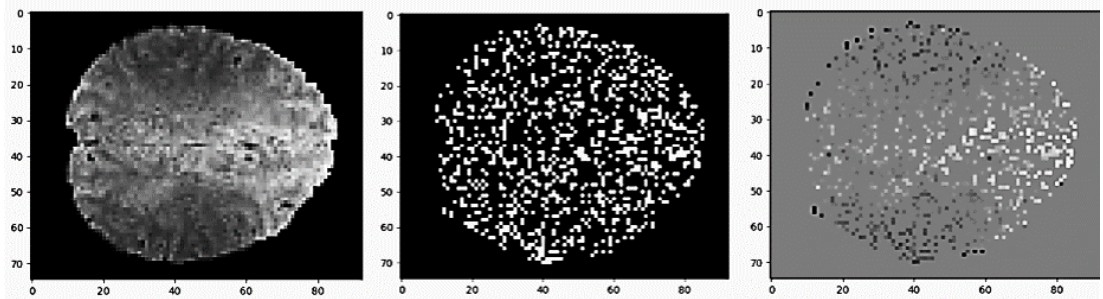

Figure 3: A representative sample of a slice (left) and its voxels that contribute to the intensity loss (right). The intensity loss emphasizes the voxels that vary the most in a given frame, by leveraging a "focused" L1 loss applied to those specific voxels. To calculate this loss, we infer a binary mask for each of the slices (middle).

The intensity loss $\mathcal{L}_1^b$ is based on an L1 loss applied to a subset of the temporally-intense voxels, which are more likely to represent a relevant BOLD signal. The intensity loss eliminates voxels associated with 80% of the values that are close to 0 were eliminated since they are typical across many frames and are therefore unlikely to represent a distinct signal. A representative slice and its voxels associated with the intensity loss can be seen in Fig. 3.

Fig. 5 presents two slices fron two representative fMRI scans (from the validation set) that were encoded and decoded by the pre-trained TFF model. As can be seen, TFF is able to preserve most of the information from the input, including the brighter areas which

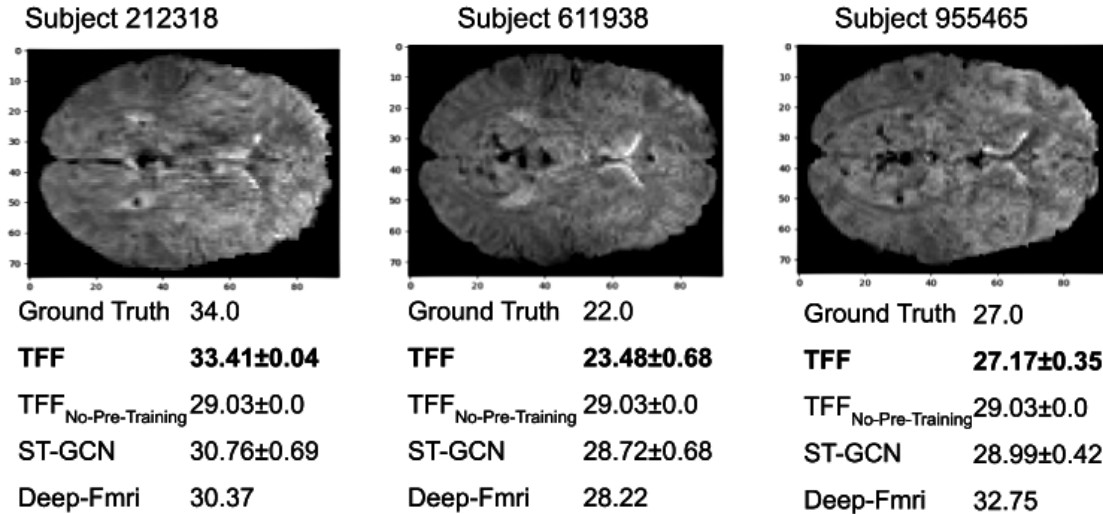

Figure 4: Three representative samples from the HCP test set, arbitrarily sliced at axial coordinate z = 40, together with the ground truth and predicted age retrieved by each model. For all models, except for the Deep-FMRI model, the average over sub-sequences is presented together with it's corresponding standard deviation. Since Deep-FMRI opertaes on full scans, each subject corresponds to one sample and a single prediction.

indicate a higher level of brain activity. This ability of the model to accurately reconstruct the volumes is an important indication for the quality of the intermediate vectors, and the sanity of the end-to-end training.

Fig.6 presents the loss values during the pre-training phase. As can be seen, the performance on the validation and training sets is highly correlated, the values of all three losses are mostly decreasing monotonically, and their values are within one order of magnitude from each other.

## Appendix B. More details about the TFF architecture

Our 3D convolutional encoder and decoder network architecture build upon the network developed by Myronenko (2018) for the purpose of brain tumor segmentation from 3D MRI data. Their architecture receives 3D MRI crops as its input, and utilizes a sequence of convolutional blocks with skip connections. The encoder output is fed into two decoder heads. The first is trained with supervision to predict segmentation. The second decoder head is trained using self-supervision to reconstruct the input volume data by employing a variational auto-encoder approach. In our work, we adopt the building blocks of the architecture, omit the decoder head that performs segmentation and modify the head that performs reconstruction by removing the variational auto-encoder mechanism. In order to support 4D fMRI data and enable the model to obtain and process temporal information, we add the transformer architecture between the encoder and the decoder.

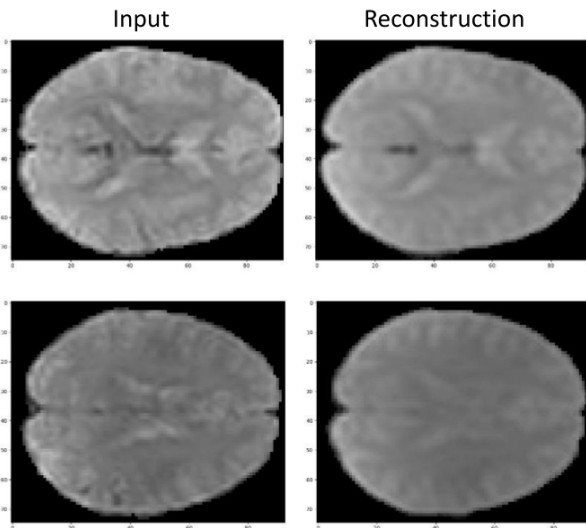

Figure 5: Two Representative slices taken from two different subjects, along with their corresponding reconstruction created by the pre-trained TFF model. The slices were arbitrarily chosen from each subject (at depths 48 and 51 respectively).

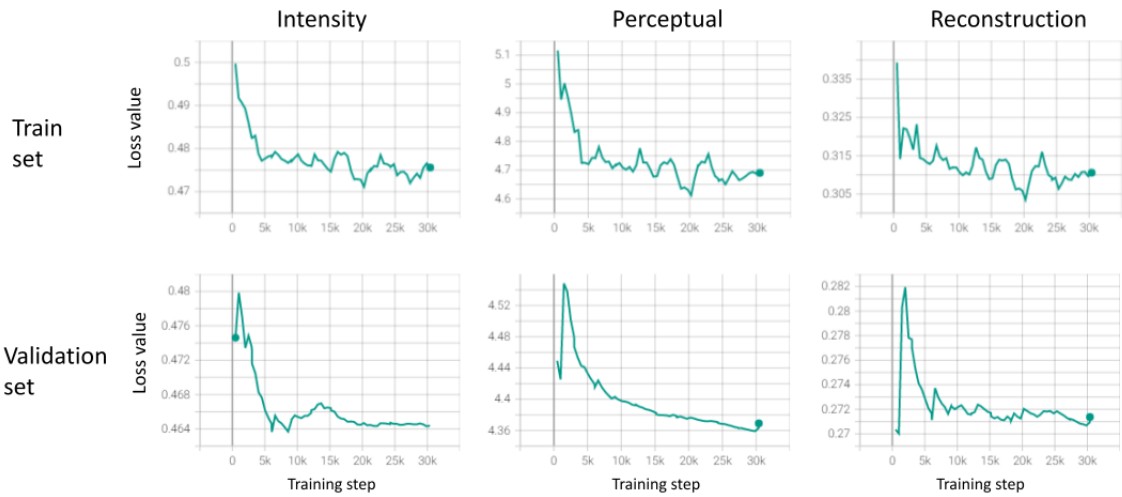

Figure 6: The loss values per training step, calculated during the pre-training phase.

Tab. 8 and Tab. 9 present the detailed architecture of the 3D encoder and decoder networks, respectively. Specifically, the encoder is composed of a sequence of four blocks, each block comprising a 3D convolutional layer, followed by a dropout layer (Srivastava et al., 2014), group normalization layer (Wu and He, 2018), ReLU, convolutional layer,

| dataset name | dataset size | scan length | $w$ | $s$ | number of samples |
|---|---|---|---|---|---|
| HCP | 1095 | 1190 | 20 | 20 | 15000 |
| CNP | 261 | 118-142 | 20 | 10 | 2233 |
| COBRE | 146 | 67-150 | 20 | 10 | 1890 |

Table 6: More details about the datasets used in our experimental section. The last column represents the number of samples during the second stage of the pre-training as well as the fine-tuning procedure.

groupNorm layer, and another ReLU. Finally, a down-sample layer is applied by utilizing a 3D convolutional layer with a stride of two. The output of the last block is then flattened, forming a 1D feature vector of size 2640. Notably, each of the elements in the flattened vector corresponds to a receptive field of 8X8X8 in the original volumetric input.

The decoder architecture applies the same number of convolutional blocks, with similar layers, except for the final down-sampling layer which is replaced by an up-sampling layer.

The implementation of the transformer architecture is based on the hugging face library[5]. Our code is attached as an additional supplementary.

During the first pre-training step, only the 3D encoder and the decoder are trained. In the second step, the transformer is integrated between the encoder and the decoder, and the entire architecture is trained to optimize the pre-training objective.

During fine-tuning, we remove the decoder, and only the encoder-transformer networks are trained, where the transformer operates on an additional CLS token that is concatenated to the beginning of each sequence. The CLS embedding is then propagated through an MLP to score a regression/classification objective.

During pre-training, the number of training samples in the first pre-training stage is equal to the number of frames in the dataset since each sample is a single 3D fMRI frame. In the second pre-training stage and during the fine-tunning, each sample is a sequence of 3D fMRI frames, the sequences are of size w and are created by using a stride s. Therefore, the number of samples during those stages depends on the length and the number of the scans. In the HCP, CNP, and COBRE datasets, it iswould be 15000, 2233, and 1890, respectively. More details can be found in Table 6.

## Appendix C. More details about the HCP dataset and evaluations

The HCP dataset (Van Essen et al., 2013) originally incorporates 1200 subjects out of which 1096 are available at the ConnectomeDB website under the category - Resting State fMRI 1 Preprocessed. The scans were pre-processed by the HCP functional pipeline available from the Connectome DB website[6]. In TFF, the HCP data is applied without any additional pre-processing, beyond the pipeline used in the Connectome DB project. The pre-processing

---

[5]https://huggingface.co/transformers/_modules/transformers/models/bert/modeling_bert.html#BertModel

[6]https://db.humanconnectome.org/app/template/Login.vm

| Model | Accuracy | BAC | AUC | Precision | Recall | F1 |
|---|---|---|---|---|---|---|
| TFF | 94.09 | 93.91 | 98.77 | 94.84 | 92.0 | 93.4 |
| TFF$_{\text{no-pre-training}}$ | 92.06 | 93.18 | 95.34 | 93.56 | 92.0 | 92.77 |
| ST-GCN | 79.81 | 82.0 | 81.36 | 82.13 | 77.0 | 79.48 |
| Deep-FMRI | 65.45 | 66.91 | 78.0 | 58.45 | 83.0 | 68.59 |

Table 7: Gender prediction results on the HCP dataset.

pipeline includes spatial artifact and distortion removal, surface generation, cross-modal registration, and alignment to standard space.

From the available 1096 subjects, we removed 1 subject which produced an error during the parcellation process. In our experiments, we split the data randomly into train, validation, and test sets, with respective sizes of 765, 110 and 220.

At the ConnectomeDB website, all subjects are listed with a respective metadata file, containing an age category (i.e. young/adult) and an associated ID number. The precise age used in our work was retrieved from the GitHub page of (Gadgil et al., 2020), which uploaded the data as a chart of subject IDs associated with accurate age.

### C.1. HCP Gender Prediction Task

Tab. 7 depicts the performance of all models, evaluated on the gender prediction task. Here, we report the accuracy, balanced accuracy, AUC, precision, recall and f1 scores for each of the models.

### C.2. HCP Age Prediction Task

Fig. 4 depicts three representative samples from the age prediction task. The figure presents the age predictions of all models for three subjects from the HCP test set, along with the ground truth and a representative slice from each of the scans.

## Appendix D. More details about the baselines

In both the Deep-fMRI and ST-GCN baselines, we performed a grid search over two parameters: the learning rate and the window size $w$. An early stopping protocol was enforced with a patient of 30 epochs, entailing that the training stopped only if 30 epochs have passed without any improvement on the validation set. In ST-GCN, we follow inference proposed by the authors. We randomly sampled sub-sequences of size $w$ from the full scan and propagated the sub-sequences through the network. The final model prediction is based on the cumulative predictions from each of the sub-sequences

## Appendix E. More details about the COBRE dataset

The COBRE dataset contains resting-state functional MRI data of 75 healthy control patients along with 72 schizophrenia patients diagnosed using the Structured Clinical Interview for DSM Disorders (fourth edition). The dataset was obtained from the Neuroimaging

| name | operations | repeats | output size |
|---|---|---|---|
| Input | Batch of fMRI volumes. Global and voxel norm aggregated on the channel dim | 1 | (2x75x93x81) |
| Conv Block 1 | Conv, Dropout | 1 | (4x75x93x81) |
| Regular Block 1 | GroupNorm, LReLU, Conv , GroupNorm, LReLU, Conv | 1 | (4x75x93x81) |
| Down Block 1 | Dropout, Conv (stride 2) | 1 | (5x8x38x47x41) |
| Regular Block 2 | GroupNorm, LReLU, Conv, GroupNorm, LReLU, Conv | 2 | (5x8x38x47x41) |
| Down Block 2 | Dropout, Conv (stride 2) | 1 | (5x16x19x24x21) |
| Regular Block 3 | GroupNorm, LReLU, Conv, GroupNorm, LReLU, Conv | 2 | (5x16x19x24x21) |
| Down Block 3 | Dropout, Conv (stride 2) | 1 | (5x32x10x12x11) |
| Regular Block 4 | GroupNorm, LReLU, Conv, GroupNorm, LReLU, Conv | 4 | (5x32x10x12x11) |
| Reduce Block | GroupNorm, LReLU, Conv | 1 | (5x2x10x12x11) |
| Flatten | Flatten | 1 | (5x2640) |

Table 8: The architecture of the 3D encoder network $\mathcal{E}$. Unless mentioned otherwise, all Convolution operations utilize a kernel of size 3, stride 1, and padding 1.

Informatics Tools and Resources Clearinghouse (NITRC) website and was published by the Center of Biomedical Research Excellence[7].

The pre-processing of this data is based on the the NIAK pipeline for rs-fMRI and includes artifact/distortion removal, band pass filtering and registration to a standard space. Subjects range by age from 18 to 65. The exclusion criteria were a history of neurological disorder, mental retardation, severe head trauma with more than 5 minutes loss of consciousness, or that of substance abuse or dependence within the last 12 months. The data were split randomly into train, validation, and test sets, with respective sizes 102, 14, and 30.

## Appendix F. More details and results for the CNP dataset

The CNP (Consortium for Neuropsychiatric Phenomics) dataset contains resting-state functional MRI data of 138 healthy control subjects along with 58 schizophrenia, 49 bipolar disorder, and 45 ADHD patients, all were assessed using the Structured Clinical Interview for DSM Disorders (fourth edition). The dataset was obtained from the openneuro initiative for shared fMRI data. Minimal preprocessing was conducted using the FMRIPREP pipeline, including coregistration, normalization, unwarping, noise component extraction, segmentation, skull stripping, etc.

All patients were assessed with the Structured Clinical Interview for the DSM (Fourth Edition). We focus on distinguishing between schizophrenia and healthy subjects, and we divided the data randomly into train, validation, and test sets, with sizes of 140, 20, and 20, respectively.

The results for comparing the convergence of TFF and TFF$_{vanilla}$ during the shared fine-tuning stage can be seen in Fig. 7. The results for the variable training set size experiment, from the main text, can be seen in Fig. 8.

---

[7]http://fcon_1000.projects.nitrc.org/indi/retro/cobre.html

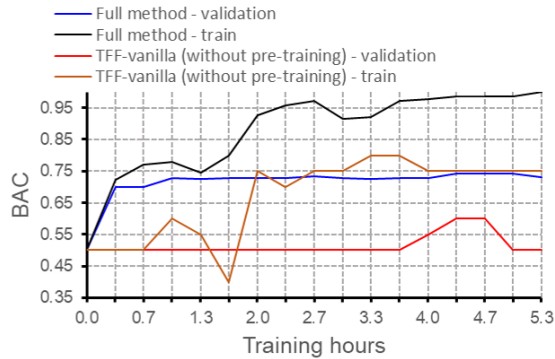
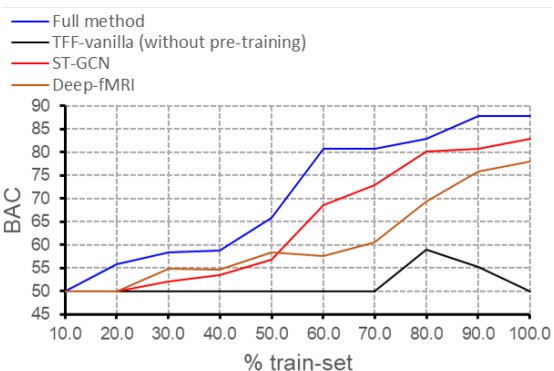

| Figure 7: BAC vs training time. | Figure 8: BAC vs. % of training set. |

| name | operations | repeats | output size |
|------|-----------|---------|-------------|
| Input | output of the encoder / attention output sequences aggregated on batch dim | 1 | (5x2640) |
| Linear Block | Linear | 1 | (5x2640) |
| Expand Dim | UnFlatten, GroupNorm, LReLU, Conv | 1 | (5x32x10x12x11) |
| Up Block 1 | Conv (kernel size 1), UpSample | 1 | (5x16x19x24x21) |
| Regular Block 1 | GroupNorm, LReLU, Conv, GroupNorm, LReLU, Conv | 1 | (5x16x19x24x21) |
| Up Block 2 | Conv (kernel size 1), UpSample | 1 | (5x8x38x47x41) |
| Regular Block 1 | GroupNorm, LReLU, Conv3d , GroupNorm, LReLU, Conv | 1 | (5x8x38x47x41) |
| Up Block 3 | Conv (kernel size 1), UpSample | 1 | (5x4x75x93x81) |
| Regular Block 1 | GroupNorm, LReLU, Conv, GroupNorm, LReLU, Conv | 1 | (5x4x75x93x81) |
| Final Block | Conv ,Conv (kernel size 1) | 1 | (5x1x75x93x81) |

Table 9: The architecture of 3D the decoder network $\mathcal{D}$.

## Appendix G.  Adopting the code for other modalities

Our code is available over GitHub[8]. It is modular and can be tweaked to support different types of images and/or data. In the code, the volume data is a simple PyTorch tensor. The data inside this tensor can be any volumetric data and can be pre-processed by various pipelines. Additionally, by tweaking the architecture of the CNN encoder-decoder, one can reshape the model to handle data of different dimensions.

In order to demonstrate the applicability of our method for other modalities, we have trained TFF on a dataset of MRI scans (instead of fMRI). Since MRI scans do not have a temporal component and are composed of single 3D volumetric data, this is an edge case for TFF, in which it is applied on sequences with a length of 1. In this scenario, the transformer model becomes a feed-forward network with a gating operator.

he OASIS-1 dataset[9], which contains 416 MRI scans of different subjects, was employed. A TFF model was pre-trained to reconstruct the acquired MRI volumes and fine-tuned to predict the age of the subjects from the acquired MRI data.

The pre-training was applied on a random split of the OASIS-1 dataset, using a train-validation-test split with 291, 62, 63 samples, respectively. Following TFF, the model was initialized with a 3D CNN auto-encoder and was trained by the first pre-training stage to reconstruct the entire MRI volumes. Each sample of the dataset is an MRI scan associated

---

[8]https://github.com/GonyRosenman/TFF

[9]https://www.oasis-brains.org/

| Model | L1 |
|---|---|
| TFF | **18.25±2.85** |
| TFF$_{vanilla}$ | 21.12± 1.22 |

Table 10: Mean±SD over five runs for the age prediction task evaluated on the OASIS-1 dataset, which composed of 416 acquired MRI scans.

| Model | L1 | L2 | NMSE ×10 |
|---|---|---|---|
| TFF | **2.74±0.04** | **11.11±0.34** | **0.14±0.01** |
| TFF$_{vanilla}$ | 3.21±0.00 | 14.15±0.03 | 0.19±0.00 |
| ST-GCN | 3.21±0.08 | 14.06±0.10 | 0.19±0.05 |
| Deep-fMRI | 3.33±0.21 | 17.12±0.10 | 0.24±0.09 |

Table 11: Mean±SD over five runs for the age prediction task evaluated on the HCP dataset. The p-value between TFF and the other three baselines is lower than 0.005.

with a different subject. The fine-tuning is initialized by the pre-trained 3D CNN encoder, with a transformer model and a regression head on top. During fine-tuning, the model is trained in an end-to-end manner, optimizing a standard regression loss to minimize the L2 distance between the model prediction and the age label of each subject.

To assess the validity of the pre-training procedure, we compared TFF to TFF$_{vanilla}$, which is the TFF baseline that does not benefit from the pre-training procedure. Importantly, both models are composed of identical TFF models, where the only difference is that TFF$_{vanilla}$ was not pre-trained to optimize a reconstruction objective.

As can be seen in Tab. 10, the proposed TFF model was able to produce a fairly good performance, where the mean average error, across five different runs, is 18.25 while the subjects' ages vary between 18-90. Compared to the TFF$_{vanilla}$, the full TFF method yields an improvement of +2.87 of accuracy, which highlights the importance of the pre-training procedure.

## Appendix H. More results for the age prediction task

We further report the mean and standard deviation (STD) values of the L1, L2, and NMSE metrics, calculated across five different runs of the age prediction task, for our method and each of the three baselines described in Section 3. Overall, the results aggregate the performance of 20 trials (5 experiments for each model). All models were trained on the age prediction task, using the HCP dataset. As can be seen in Tab.11, TFF outperforms the other alternatives by a sizeable margin (the error bars do not overlap) while producing stable results over multiple runs (relatively low standard deviation). The p-value between TFF and the other three baselines is lower than 0.005 in all cases.

