# OpenReview forum: "Self-Supervised Transformers for fMRI representation"
_MIDL.io/2022/Conference — MIDL 2022_

### Official Review · Reviewer_VXqt · 2022-01-24

**Confidence:** 4
**Preliminary Rating:** 3
**Recommendation:** Poster

**Summary:**

The authors proposed a transformer-based framework to analyze functional MRI data in particular 4D fMRI
volume data which is trained in an end-to-end manner. A set of experiments was made on different tasks such as age and gender prediction, and schizophrenia recognition. In addition, a comparison with state-of-the-art methods were performed and the model was evaluated on three datasets.

**Strengths:**

1- The main advantage is the tackle of an interesting problem in the medical field that is fMRI data analysis which can be used for early disease diagnosis and,
2- the novelty lies in proposing a transformer-based approach in order to solve such a problem.
3- The paper is well written, results are clear, and show the outperformance of the proposed model compared to the existing works.

**Weaknesses:**

1- Lack of explanation of the transformer network and the reason for choosing it instead of any other DL network as well as the self-supervision learning strategy.
2- Unclear choice of a graph-based method (ST-GCN) for comparison while the data treated in the paper is an image.
3- Lack of discussion of the weakness of the proposed model.
4- Lack of explaining the novelty of the method in the introduction section.

**Deanonymize Review:**

no

**Final Rating After The Rebuttal:**

4: Weak Accept

**Justification Of The Final Rating:**

The authors performed additional experiments to show the impact of their model when using different types of images and improved the paper by adding some parts in the introduction and the appendix. I see that the second version of the paper is good to be published in MIDL.

**Paper Type:**

methodological development

**Questions To Address In The Rebuttal:**

How is the model performing on different types of images? Have the authors created a generic code that can be used for different medical images? In which case the model can fail? Why transformer? What is unique in such a model? These points need to be well-argued in the paper.

**Special Issue:**

no

---

### Official Review · Reviewer_Fi7M · 2022-01-27

**Confidence:** 5
**Preliminary Rating:** 4
**Recommendation:** Poster

**Summary:**

This work proposed the transformer-based self-supervised learning pipeline for fMRI representation learning. Specifically, fMRI volumes are fed into a transformer for the pre-text task of reconstruction. In such a pre-training stage, a 3D encoder with conv layers, a transformer, and a 3D CNN decoder are trained. In the fine tuning stage, the 3D encoder and transformer blocks will be used to predict the final task (i.e., classification) by fine-tuning. The proposed method was validated on HCP datasets, COBRE, and CNP datasets.



**Strengths:**

1. The overall application of transformer and SSL on fMRI analysis is new.
2. The paper is easy to follow and related work is well-addressed
3. The proposed method is validated on multiple datasets and show promising results

**Weaknesses:**

1. Although the application is new of using transformer, but the rationality why transformer should be added into the pipeline is not clear.
2. As fMRI is very noisy and temporal information is important, i am a little bit unconfident doing volume-level reconstruction for pre-training is the best choice.
3. Given the limited number of training samples, it will be great if the authors could justify the performance on pre-training. Is it possible to visualize the reconstruction results?
4. It will be helpful to report cross-validation results.

**Deanonymize Review:**

no

**Final Rating After The Rebuttal:**

4: Weak Accept

**Justification Of The Final Rating:**

I appreciate authors' rebuttal. Most of my concerns are addressed. If the final version could improve the clarity, especially on the rationality, this is a good work to be accepted. Thus I will give week accept to this submission.

**Paper Type:**

methodological development

**Questions To Address In The Rebuttal:**

1. Highlight the motivation why the pretraining will benefit fMRI analysis
2. Highlight the rationality of choosing transformer
3. Explain why reconstructing volume is a reasonable SSL task
4. Are different loss terms at the same scale?
5. Specify the number of training samples at each statge
6. (if possible) report std on multi-trials

**Special Issue:**

no

---

### Official Review · Reviewer_Vg11 · 2022-01-27

**Confidence:** 3
**Preliminary Rating:** 3
**Recommendation:** Poster

**Summary:**

In the paper the authors present a Transformer framework for the analysis of functional Magnetic Resonance Imaging (fMRI) data. It employs a two-phase training approach, where first the model is trained to reconstruct 3D volume data in a supervised fashion and then the pre-trained model is fine-tuned on specific tasks, utilizing ground truth labels. The results show state-of-the-art performance on a variety of fMRI tasks, including age and gender prediction, as well as schizophrenia recognition.

**Strengths:**

There are strengths to highlight in this paper:

1) Using a transformer-based architecture is a novel methodological application

2) The ability to cover the entire 4D fMRI volume data is very interesting, since parcellations always have inherent limitations

3) The code is fully avaiableble and public datasets were used which should enable reproducing the work in its entirety.



**Weaknesses:**

The clarity of the paper, especially on the neuroscientific part needs to be improved.

1) In the main article there are no details what so ever on any preprocessing of the fMRI data. Besides parcellating, there are a numbers of steps, e.g. motion correction, de-noising that are normally standard preprovcessing for fMRI data. One needs to dig into page 16 of the appendix and there the preprocessing is still not detailed. So what kind of 4D fMRI volumes are you feeding into the algorithm?

2) The first 5 pages are spent on the methods and the details, especially why normalization of the signal needs to be performed in certain ways, are not very clear.

3) The regression and prediction tasks are standard tasks, but in the descriptions of the datasets, it's partially not even clear what kind of fMRI data is used. Only for HCP it's clearly stated that you are using resting state fMRI data, for the COBRE and CNP it just states fmRI data. That needs to be clarified. Also why are those tasks even sensible to do?

4) Especially your comparisons to baselines are fine when you just want to do method development, but you do not compare your quality of e.g. disease status prediction to the general standard based on classical clinicla valuables? Does it even make sense to acquire expensive fMRI data and to train a heavy machine learning model for the clinic or cna basic variables predict disease status just as well? I am lacking the perspective of why you ahve chosen these tasks and how your work is contributing to the developments in neuroscience besides the data just being a nice plyaing field for method development.


**Deanonymize Review:**

yes

**Final Rating After The Rebuttal:**

4: Weak Accept

**Justification Of The Final Rating:**

I thank the authors for their answers to me as well as to the other reviewers. They have provided a thorough revision that ha improved the clarity on the neuroscientific part tremendously. I therefore improve my rating slightly to a weak accept.

**Paper Type:**

methodological development

**Questions To Address In The Rebuttal:**

I would like the authors to address my questions on the neuroscientific details, such as the pre-processing, the data descriptions and especially the reasoning og task choices for validating the model on real data.

**Special Issue:**

no

---

### Meta-Review · Area_Chair_b4Sy · 2022-02-21

**Recommendation:** Accept (Poster)
**Confidence:** 4

**Metareview:**

While there were several concerns upon initial submission, including clarity of experimental details and missing rationality for transformer model for fMRI and self-supervision strategy, upon revision reviewers all leaned toward acceptance, with paper strengths including: 1) new application of transformers to fMRI analysis, 2) good validation on multiple datasets, 3) well-written paper.

---

### Decision · Program_Chairs · 2022-02-28

Accept